**Investigation**

# Sex bias in iron sequestration by transferrin 1 modulates sexually dimorphic infection outcomes in *Drosophila melanogaster*

Alexandra Hrdina (iD) ,[1,2] Igor Iatsenko (iD) [1,*]

[1]Research Group Genetics of Host–Microbe Interactions, Max Planck Institute for Infection Biology, Berlin 10117, Germany
[2]Humboldt-Universität zu Berlin, Faculty of Life Sciences, 10099 Berlin, Germany

*Corresponding author: Research Group Genetics of Host–Microbe Interactions, Max Planck Institute for Infection Biology, Charitéplatz 1, Berlin 10117, Germany.
Email: iatsenko@mpiib-berlin.mpg.de

Host sexual dimorphism in the outcome of infections is a ubiquitous phenomenon across taxa. However, the immunological differences between males and females and the mechanisms underlying them remain poorly characterized. Here, we used *Drosophila melanogaster* to test the hypothesis that sex differences in nutritional immunity, particularly iron sequestration, contribute to sexual dimorphism in infection outcome. Using the natural *Drosophila* pathogen *Providencia alcalifaciens*, which is controlled by host-mediated iron sequestration, we established an infection model in which males demonstrate increased resistance. Leveraging this model, we showed that males exhibit higher basal and infection-induced expression levels of *Transferrin 1* (*Tsf1*)—an iron transporter mediating iron sequestration during infection. Consequently, males contained lower iron levels in the hemolymph compared to females. Importantly, sex differences in iron content and in survival after *P. alcalifaciens* infection were abolished in *Tsf1* mutants, demonstrating that Tsf1 mediates sexual dimorphism in iron sequestration and susceptibility to *P. alcalifaciens* infection. Finally, we found that the Toll pathway mediates sex differences in *Tsf1* expression and susceptibility to infection. Altogether, our study demonstrates that Tsf1-mediated iron sequestration differs between male and female *D. melanogaster*, thereby identifying nutritional immunity as a determinant of sexual dimorphism in the outcome of infection.

Keywords: *Drosophila melanogaster*; nutritional immunity; iron sequestration; transferrin; sexual dimorphism; infection; immunity

## Introduction

Iron plays an essential role in numerous biochemical processes and is therefore vital for the survival of almost all life forms. It can be found as a cofactor in a wide variety of proteins, mediating processes such as oxygen transport, gene regulation, and metabolism. The natural ability of iron to cycle between 2 oxidation states, $Fe^{2+}$ and $Fe^{3+}$, renders iron an important redox catalyst, which is highlighted by the fact that iron is the most abundant redox metal in biological systems (Andreini et al. 2008; Hood and Skaar 2012; Cassat and Skaar 2013). Nonetheless, iron levels need to be tightly controlled as free iron and reactive oxygen species (ROS) catalyze a Fenton reaction, generating hydroxyl radicals that damage cells (Cassat and Skaar 2013; Gomes et al. 2023). Subsequently, iron does not occur in its free form within the organism but is instead bound to iron-binding molecules and proteins, such as heme, transferrin, and lactoferrin (Kaplan and Ward 2013; Cao et al. 2023).

The importance of iron in sustaining life raises issues at the interface between hosts and pathogens during infection. While hosts acquire iron via dietary intake or by modifying the regulation of storage and use, pathogens rely on iron to be present in their immediate environment (Hood and Skaar 2012; Golonka et al. 2019). Host tissues present an iron-limited environment as

the present iron is bound to iron-binding proteins and therefore unavailable to the pathogen. Furthermore, hosts will further restrict pathogen iron-access by sequestering iron from infected tissue. Immune responses, such as these, which involve transition metals, are generally termed nutritional immunity (Lopez and Skaar 2018; Hrdina and Iatsenko 2022; Murdoch and Skaar 2022). Conversely, pathogens have developed sophisticated strategies to overcome these defenses in order to successfully colonize. One prominent strategy is the production of siderophores, iron-chelating molecules that have a higher affinity to iron than the iron-binding proteins produced by the host (Hider and Kong 2010; Golonka et al. 2019; Kramer et al. 2019; Helmann 2025).

*Drosophila melanogaster* is a widely appreciated model organism to study host–pathogen interactions due to its genetic amenability (Westlake et al. 2024). Curiously, nutritional immune responses have been understudied in the fruit fly (Missirlis 2021; Hrdina and Iatsenko 2022). The best investigated immune response involving transition metals in *Drosophila* is iron sequestration mediated by the iron transporter Transferrin-1 (Tsf1) (Xiao et al. 2019). Following infection, *Tsf1* expression has been shown to be induced via Toll and IMD signaling and is responsible for removing iron from the hemolymph and shuttling it into fat body cells. Tsf1-deficient flies are more susceptible to multiple pathogens

like *Pseudomonas* and *Providencia* bacteria as well as *Mucorales* fungi, highlighting the importance of iron in the defense against infection (Iatsenko et al. 2020; Shaka et al. 2022; Hrdina et al. 2024).

Biological sex has a great influence on the physiology of an organism (Millington and Rideout 2018; Dähn and Wagner 2025). Sex differences are most obvious in anatomy and behavior but have also been reported in gene expression, epigenetics, and hormone levels (Ratnu et al. 2017; Snell and Turner 2018; Dähn and Wagner 2025). Consequently, males and females respond differently to a variety of stressors, including infection and inflammation (Khan and Graze 2024; Rubinić et al. 2025). Even though sexual dimorphism has been reported in immunity (Klein and Flanagan 2016; Takahashi and Iwasaki 2021), research on this topic has been severely impacted by missing reports of biological sex and a preference for unisexual studies within the field (Belmonte et al. 2020). Furthermore, the study of sexual dimorphism in immunity is made more complex by the fact that it is also dependent on the pathogen (Duneau et al. 2017; Belmonte et al. 2020). Hence, the mechanisms underlying sex differences in infection outcome remain poorly characterized.

It has previously been demonstrated that male and female flies differ in survival after pathogen challenge (Belmonte et al. 2020). Male flies were more resistant against infection with *Providencia alcalifaciens* and *Providencia rettgeri*, while females were less susceptible to *Staphylococcus aureus*. Moreover, it was shown that the regulation of the Toll pathway underlies sexual dimorphism and is more strongly induced in male flies compared to female flies (Duneau et al. 2017). Another study demonstrated that Tsf1 levels in the hemolymph are significantly higher in adult male flies (22 μM) compared to female flies (4.4 μM) (Weber et al. 2022). These results introduce the possibility of attributing differential disease outcomes to differences in iron levels between male and female flies. However, a direct link between iron sequestration and the sexually dimorphic survival of flies following infection has not yet been established. In this study we investigate whether iron sequestration is sexually dimorphic and if it contributes to sex differences in survival outcome.

Here, we found evidence that the sexually dimorphic survival is linked to lower iron levels in male hemolymph due to higher *Tsf1* expression in males compared to females. These differences are likely caused by the sexually dimorphic activation of the Toll pathway. Altogether, we show that iron sequestration, a nutritional immune response, is sexually dimorphic and influences the response to infection between the sexes.

## Materials and methods
### *Drosophila* stocks and rearing
The following *Drosophila* stocks used in this study were described previously: DrosDel $w^{1118}$ iso; Oregon R; *Relish*$^{E20}$ iso; *spz*$^{RM7}$ iso; *Tsf1*$^{JP94}$ iso; *PPO1*$^4$,2$^4$ iso; *c564-GAL4; UAS-Tsf1; UAS-CD8-GFP* (Dudzic et al. 2019; Iatsenko et al. 2020; Marra et al. 2021; Shaka et al. 2022; Arias-Rojas et al. 2023; Rubinić et al. 2025). The stocks were routinely maintained at 25 °C with 12/12 h dark/light cycles on a standard cornmeal-agar medium: 3.72 g agar, 35.28 g cornmeal, 35.28 g inactivated dried yeast, 16 mL of a 10% solution of methylparaben in 85% ethanol, 36 mL fruit juice, 2.9 mL 99% propionic acid for 600 mL. Fresh food was prepared weekly to avoid desiccation. Flies were flipped to new vials with fresh food every 3 to 4 d to grow new generations. Flies used in the experiments were 3 to 7 d old, and wild type and mutants as well as males and females were age-matched. Male and female flies were kept in the same vial and separated only shortly before experiments (<24 h) to ensure similar conditions. Virgin flies were collected shortly after eclosion male and female flies were kept in separate vials until the experiment.

### Pathogen strains and survival experiments
In this study we used *P. alcalifaciens* DSM30120 that was obtained from the German Collection of Microorganisms and Cell Cultures (DSMZ) and *P. rettgeri* (Galac and Lazzaro 2011). The strains were grown in LB media (Invitrogen) overnight at 37 °C in a shaking incubator. The culture was pelleted by centrifugation to remove the media and diluted to the desired optical density (OD600 = 0.25, 0.5, 1, 2, 3) with sterile PBS. If not otherwise indicated, flies were infected with OD600 = 2. To infect flies, a 0.15 mm minutien pin (Fine Science Tools) mounted on a metal holder was dipped into the diluted bacterial solution and poked into the thorax of a CO2 anesthetized fly. Infected flies were maintained in vials (20 flies per vial) with food at 25 °C overnight and moved to 29 °C around 15 to 18 h post infection. Surviving flies were counted at regular intervals. One to 3 vials per genotype and sex were infected per experiment, and the survivals were repeated at least 2 times using independent generations of flies. For the survival experiments that involved the prefeeding of flies with chemical compounds or sucrose, flies were fed for 24 h with a mix of 2.5% sucrose + 5 mM ferric ammonium citrate (FAC) or 2.5% sucrose (control group). Then 150 μL of the respective solution was applied on top of a filter disc covering the fly food. Flies were flipped into fresh vials without the filter after infection.

### Quantification of pathogen load
For bacterial counts, flies were infected with *P. alcalifaciens* as described above and kept at 25 °C. The number of bacteria was determined as follows at 0, 2, 6, 10, and 16 h postinfection: Flies were surface sterilized in 70% ethanol and 3× for 5 s and then washed in sterile PBS 1× for 5 s. Flies (1 fly per replicate) were homogenized in 500 μL of sterile PBS for 30 s at 7,200 rpm using a Precellys 24 instrument (Bertin Technologies, France). Serial 10-fold dilutions were made and plated on LB culture medium. The plates were left to dry and incubated overnight at 37 °C. Colonies were counted, and CFU were calculated as described previously (Hrdina et al. 2024).

### RT-qPCR
For quantification of messenger RNA, whole flies ($n = 10$) were collected at the indicated time points. Total RNA was isolated using TRIzol reagent (Invitrogen) and dissolved in RNase-free water. In addition, 500 ng of total RNA was then reverse transcribed in 10 μL reactions using PrimeScript RT (TAKARA) and random hexamer primers. The qPCR was performed on a LightCycler 480 (Roche) using the SYBR Select Master Mix from Applied Biosystems. RP49 was used as a housekeeping gene for normalization. Primer sequences were published previously (Hrdina et al. 2024). At least 2 biological replicates were collected per experiment, and experiments were repeated at least 2 times with samples from flies from independent generations.

### Hemolymph extraction and colorimetric iron measurement
To extract hemolymph, about 75 to 100 flies were anesthetized and placed on 2 10 μm filters inside of an empty Mobicol spin column (MoBiTec). Glass beads were added on top of the flies, and columns were centrifuged for 10 min at 4 °C, 5,000 × *g*. The hemolymph was collected in 50 μL of protease inhibitor cocktail (Sigma-Aldrich, Catalog #11697498001, 1 tablet in 4 mL PBS).

Then each sample was diluted in a 1:10 ratio, and the amount of protein was measured using the Pierce BCA Protein Assay Kit (Thermo Fisher Scientific) according to the manufacturer's protocol. Iron was quantified using a colorimetric assay as previously described in Xiao et al. (2019). Iron content was measured from hemolymph aliquots containing 120 μg of total protein. Protein samples (filled up to 50 μL with protease inhibitor cocktail) were hydrolyzed with 11 μL of 32% Hydrochloric acid (VWR chemicals) under heating conditions (95 °C) for 20 min and centrifuged for 10 min at 20 °C, $16,000 \times g$. Then 18 μL of 75 mM ascorbate (Sigma-Aldrich) was added to 45 μL of supernatant followed by 18 μL of 10 mM ferrozine (Sigma-Aldrich) and 36 μL saturated ammonium acetate (Chem-Lab NV). Absorbance was measured at 562 nm using an Infinite 200 Pro plate reader (Tecan). Quantification was performed using a standard curve generated with serial dilutions of a 10 mM FAC stock dilution. Flies that were prefed with FAC or sucrose were washed 2× in 1 mL PBS to remove potential traces of the chemicals prior to hemolymph extraction. Replicates were collected from independent generations of flies.

## Quantification and statistical analysis

Data representation and statistical analysis were performed using the GraphPad Prism 10 software. Each experiment was repeated independently a minimum of 3 times (unless otherwise indicated), and error bars represent the standard deviation (SD) of replicate experiments. The survival graphs show the cumulative survival analyzed using the Cox-proportional hazard model. Other data were analyzed using two-way ANOVA. Where multiple comparisons were necessary, appropriate Tukey or Šídák's post hoc tests were applied. P-values above 0.05 were considered nonsignificant. Data used to prepare graphs are summarized in Supplementary Tables 1 to 6.

## Results

### Sexual dimorphism in *Drosophila* resistance to *P. alcalifaciens* infection

To address the role of iron sequestration in sexually dimorphic infection outcome, we first wanted to establish a suitable model. We decided to use *P. alcalifaciens*—a pathogen that has previously been shown to be more virulent in female flies (Duneau et al. 2017). Importantly, Tsf1-mediated iron sequestration is an effective host defense mechanism against *P. alcalifaciens* (Shaka et al. 2022). Hence, this pathogen can be used to investigate the link between sexual dimorphism in iron sequestration and infection susceptibility. We confirmed increased susceptibility of female flies to *P. alcalifaciens* infection using *w1118* iso and *Oregon R* flies (Fig. 1a and Supplementary Fig. 1a). This phenotype was reproducible across different pathogen doses (Supplementary Fig. 1b to e) and was observed with another *Providencia* sp.—*P. rettgeri* (Fig. 1b). Also, increased susceptibility of females was present in melanization-deficient $PPO1^4,2^4$ mutant (Supplementary Fig. 1f), suggesting that melanization does not play a role in the dimorphism. Given that mating and reproduction have been shown to suppress immune responses in *D. melanogaster* females (Schwenke et al. 2016; Gordon et al. 2025), we tested whether increased susceptibility of females is a consequence of the immunosuppressive effect of reproduction. As shown in Fig. 1c, sex differences in survival were still observed in virgin flies, indicating that the higher susceptibility of female flies is not due to a higher reproduction-immunity tradeoff than in males. Consistent with an increased susceptibility to infection, female flies showed a trend toward higher pathogen loads compared to male flies (Fig. 1d). Among all the timepoints tested, we observed statistically significant differences in pathogen load only at an early timepoint (6 h postinfection), suggesting that the pathogen proliferates faster in female hosts at initial stages of infection.

### Sex-biased *Tsf1* expression leads to sexually dimorphic iron sequestration

To investigate the role of Tsf1 in the observed sex differences in infection susceptibility, we first compared *Tsf1* expression between males and females using *Tsf1* expression data extracted from FlyAtlas 2 (Krause et al. 2022). As shown in Fig. 2a, *Tsf1* expression was significantly higher in males than in females in the whole body and in most of the tissues tested. Using qPCR, we confirmed a significantly higher basal expression level of *Tsf1* in males in $w^{1118}$ iso and *Oregon R* backgrounds (Fig. 2b, Supplementary Fig. 2a). While *P. alcalifaciens* infection-induced *Tsf1* expression in both sexes, the expression level remained significantly higher in male flies at both time points post infection (Fig. 2b, Supplementary Fig. 2a).

Given the prominent role of Tsf1 in infection-induced iron sequestration, we tested whether sex-biased survival correlates with sex differences in iron sequestration. We measured the iron content in the hemolymph of uninfected male and female flies, as well as infected flies with *P. alcalifaciens*, using a ferrozine assay. Iron levels were significantly ($P = 0.0031$, two-way ANOVA) higher in female flies than in male flies in $w^{1118}$ iso and *Oregon R* backgrounds under uninfected conditions (Fig. 2c, Supplementary Fig. 2b). We have not detected any significant differences in iron levels between males and females at various timepoints after *P. alcalifaciens* infection, although there was a trend toward higher iron load in female flies (Fig. 2c). These results suggest that basal differences between sexes in iron content prior to infection might contribute to sexual dimorphism in survival. To test this possibility, we prefed flies with ferric ammonium citrate (FAC) prior to infection to saturate the iron load. FAC feeding led to a significant increase in hemolymph iron levels in both sexes. However, females accumulated more iron than males, while males after FAC feeding reached iron levels of sucrose-fed females (Fig. 2d). FAC feeding increased male susceptibility to *P. alcalifaciens* infection, while there was no significant effect of FAC on female survival (Fig. 2e). Importantly, in contrast to control sucrose-fed flies, FAC-fed flies exhibited no sex differences in survival to *P. alcalifaciens* infection (Fig. 2e). Hence, increasing iron levels through FAC feeding increases hemolymph iron levels, makes males more susceptible to infection, and eliminates sex differences in survival.

### Tsf1 mediates sexual dimorphism in survival and iron sequestration

To investigate whether Tsf1 mediates sex differences in infection susceptibility and iron content, we assessed the survival of a *Tsf1* mutant to *P. alcalifaciens* infection. As expected from our previous work (Shaka et al. 2022), *Tsf1* mutants were more susceptible to *P. alcalifaciens* infection (Fig. 3a). Notably, in contrast to wild-type flies, *Tsf1* mutants did not show significant differences between males and females in survival (Fig. 3a), irrespective of the infection dose we used (Supplementary Fig. 3a to c). Similar results were observed with *P. rettgeri* (Fig. 3b). Therefore, Tsf1 is an important mediator of sexual dimorphism in survival to some pathogens. Consistent with the reduced survival, we detected higher pathogen loads in *Tsf1* mutant males and females as compared to wild-type flies (Fig. 3c). As anticipated from our previous analysis of wild-type flies (Fig. 1d), *P. alcalifaciens* reached a higher titer

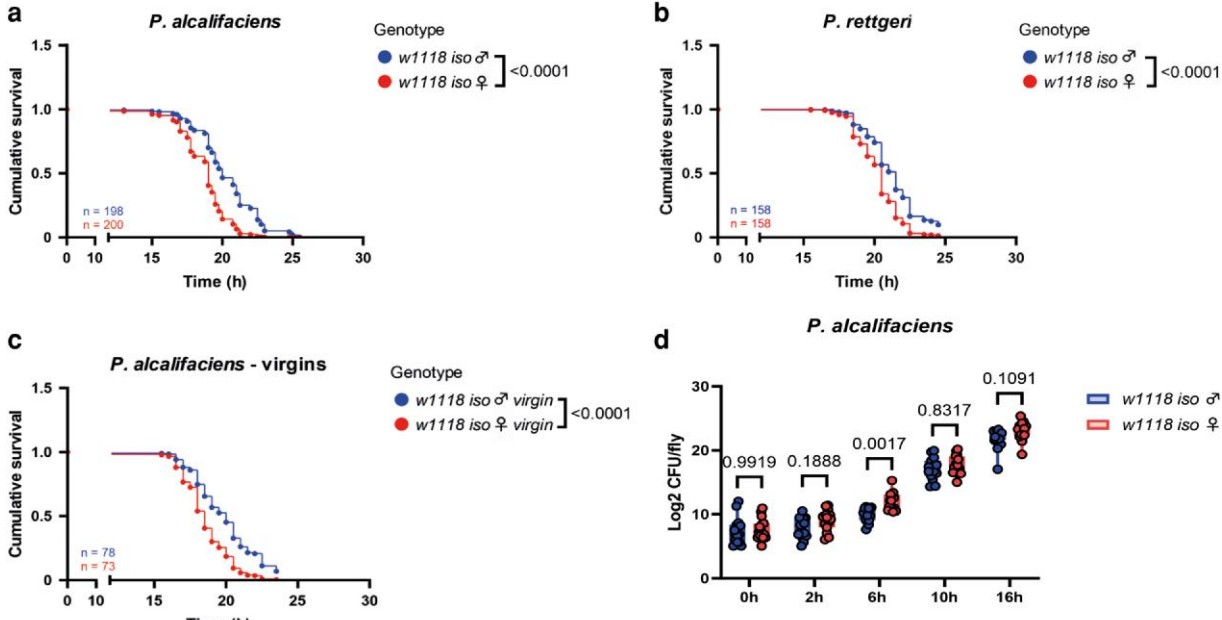

**Fig. 1.** Male flies are more resistant to infection with *Providencia* sp. than female flies. a to c) Cumulative survival of mated (a) or virgin (c) adult flies infected with *P. alcalifaciens* and mated adult flies infected with *P. rettgeri* (b). *n* indicates the total number of flies used in the experiments per genotype. d) *P. alcalifaciens* load at 0, 2, 6, 10, and 16 h postinfection. For CFU counts, 1 dot represents the pathogen load of 1 infected fly. Results are shown as mean and SD of 3 independent experiments. Data were analyzed using two-way ANOVA, Šídák's multiple comparisons test.

in wild-type females than in males at early stages of infection (6 h). The sexual dimorphism in *P. alcalifaciens* load that was observed in wild-type flies was not present in the *Tsf1* mutant, consistent with the lack of sex differences in the survival of this mutant (Fig. 3c).

To reinforce the role of Tsf1 in mediating sexual dimorphism, we observed the survival of flies with *Tsf1* overexpression in the fat body after infection with *P. alcalifaciens*. In contrast to the control line overexpressing GFP, the *Tsf1* overexpressing line showed no significant differences between sexes in survival (Fig. 3d). This loss of sex differences could be attributed to an increased survival of female flies with *Tsf1* overexpression, as such overexpression had no significant effect on the males' survival (Fig. 3d).

Finally, measuring iron levels revealed that *Tsf1* mutant flies contained more iron in the hemolymph compared to wild-type flies (Fig. 3e). Also, *Tsf1* mutant females tended to have higher iron levels than males, especially at 15 h. However, in contrast to wild-type flies which showed sexual dimorphism in basal iron load, no significant differences between sexes were identified in the *Tsf1* mutant flies (Fig. 3e), suggesting that Tsf1 mediates sexual dimorphism in the hemolymph iron content.

## Sex differences in the Toll pathway activation determine sexual dimorphism in *Tsf1* expression and fly survival

Next, we wanted to identify the signaling pathway controlling the sex-biased *Tsf1* expression. Given that *Tsf1* expression can be controlled by the Imd or Toll pathway depending on the pathogen (Iatsenko et al. 2020), we investigated whether any of these pathways contribute to the sexually dimorphic expression of *Tsf1*.

To test the role of the Imd pathway, we measured *Tsf1* expression in a *Relish* mutant. As shown in Fig. 4a, *Tsf1* expression after infection was significantly lower in the *Relish* mutant as compared to wild-type flies in both sexes. However, the sexual dimorphism in *Tsf1* expression was still observed in *Relish* mutant flies, as illustrated by the significant differences in the expression levels between males and

females after infection. These results demonstrate that while the Imd pathway is required for the full *Tsf1* induction after infection, it does not mediate sex differences in *Tsf1* expression. Next, we investigated the contribution of the Toll pathway using a *spz* mutant and found that in *spz* mutant flies, sex differences in *Tsf1* expression were not present (Fig. 4b). However, *Tsf1* expression was induced in *spz* mutant flies by infection to the same degree as in wild-type flies (Fig. 4b). Therefore, the Toll pathway is not required for *P. alcalifaciens*-induced *Tsf1* expression, but it mediates sex differences in *Tsf1* expression. Consistent with the expression data, we found that *Relish* mutants exhibited sex differences in survival with females being more susceptible (Fig. 4c). In *spz* mutants, we did not find significant differences in the survival between males and females (Fig. 4d), suggesting that the Toll pathway mediates basal sex differences in *Tsf1* expression and susceptibility to infection. Given that the sensing of microbial proteases by the *Drosophila* serine protease Persephone (Psh) upstream of *spz* was previously shown to be responsible for the sexual dimorphism in Toll pathway activation and susceptibility to *P. rettgeri* infection (Duneau et al. 2017), we investigated the contribution of the Persephone branch to the sex differences in *Tsf1* expression. Considering that Psh regulates Toll activation by microbial proteases redundantly with Hayan (Dudzic et al. 2019; Shan et al. 2023), we used a double *Psh-Hayan* mutant. We assessed the activation of the Toll pathway by *P. alcalifaciens* in *Psh-Hayan* mutant flies by measuring the expression of *Drosomycin* —an antimicrobial peptide gene controlled by Toll. Consistent with previous work (Duneau et al. 2017), *Drs* was induced stronger in wild-type males than in females (Fig. 4e), confirming stronger activation of the Toll pathway in males. Importantly, sex differences in *Drs* induction were lost in the *Psh-Hayan* mutant (Fig. 4e). Sex differences in survival to *P. alcalifaciens* infection were also abolished in *Psh-Hayan* mutant flies (Fig. 4f). These results confirm previous findings obtained with *psh* mutant flies that pathogen sensing by the Psh branch leads to sexually biased activation of the Toll pathway and consequent differences in susceptibility to infection. When we measured *Tsf1* expression, we found that sex differences were still

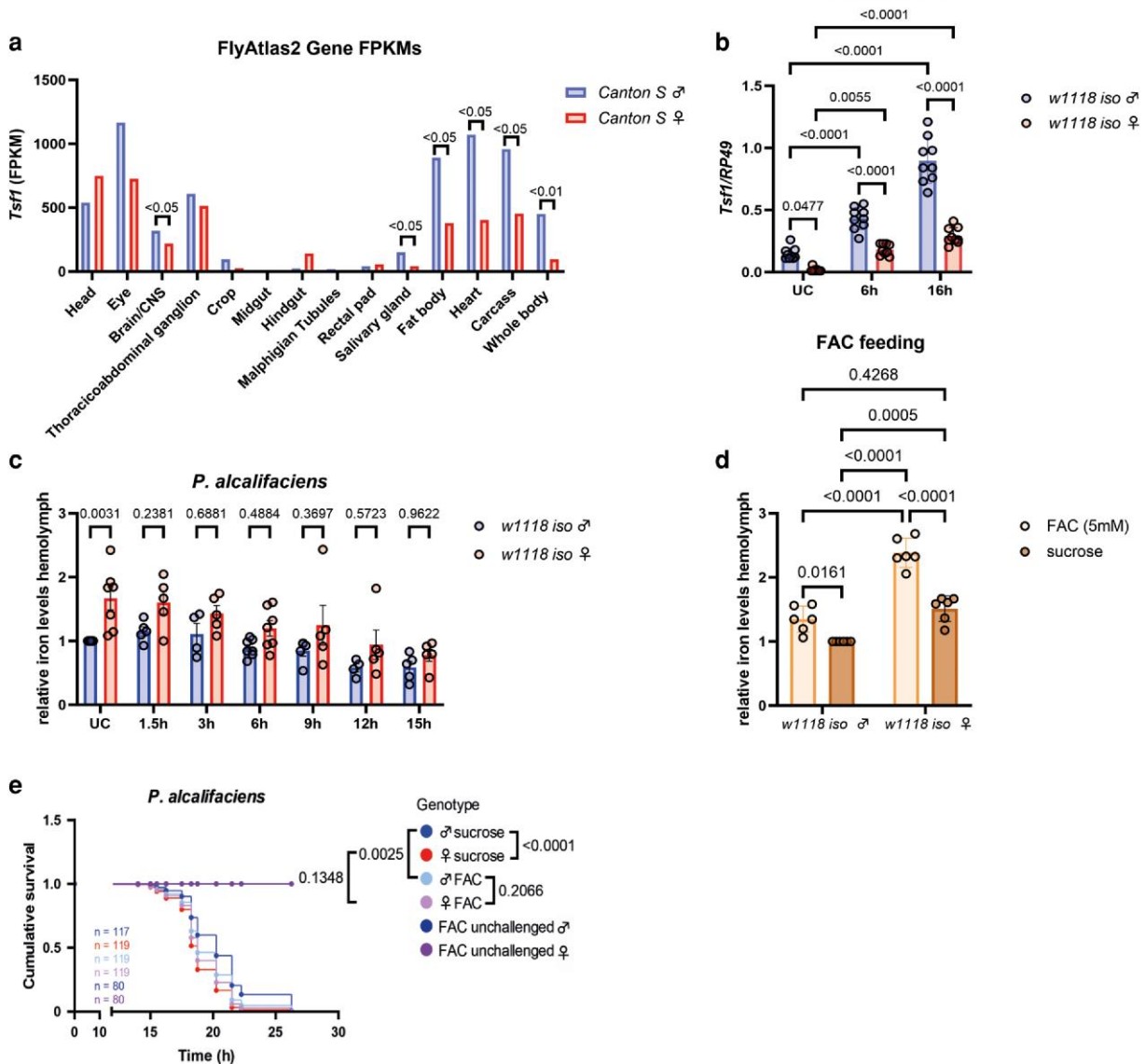

**Fig. 2.** Male flies show higher *Tsf1* levels and lower basal iron content in the hemolymph than female flies. a) Expression data of *Tsf1* in whole flies and specific tissues taken from FlyAtlas 2. b) RT-qPCR measuring *Tsf1* expression in unchallenged (UC) or flies infected with *P. alcalifaciens* 6 and 16 h postinfection, respectively. One dot represents expression levels from 10 pooled flies. The mean and SD of 3 independent experiments are shown. Data were analyzed using two-way ANOVA, Tukey's multiple comparisons test. c) Relative iron levels in the hemolymph of unchallenged (UC) flies or after infection with *P. alcalifaciens* after indicated timepoints. The iron content was measured using the ferrozine assay. One dot represents the iron levels of a pool of 75+ flies. Data shown as relative values compared to iron levels in unchallenged male flies. The mean and SD of at least 3 independent experiments are shown. Data were analyzed using two-way ANOVA and Šídák's multiple comparisons test. d) Relative iron levels in the hemolymph of flies prefed with FAC or sucrose as control for 24 h. The iron content was measured using the ferrozine assay, and 1 dot represents iron levels of a pool of 75+ flies. Data shown as relative values compared to iron levels in sucrose-fed males. Data were analyzed using two-way ANOVA and Tukey's multiple comparisons test. e) Cumulative survival of *w1118 iso* flies that were previously fed with FAC or sucrose as a control for 24 h. Unchallenged flies were only fed with FAC but not infected. *n* indicates the total number of flies used in the experiments per genotype.

present in the *Psh-Hayan* mutant under both uninfected and infected conditions (Fig. 4g). Hence, while *spz* mediates basal differences in *Tsf1* expression between sexes, there is a *Psh-Hayan*-independent mechanism of *spz* activation that is relevant for basal sex bias in *Tsf1* regulation.

## Discussion

Across taxa, understanding the differences between males and females in infection outcomes remains a significant unresolved question. In this study, we showed that sex differences in

nutritional immunity, specifically iron sequestration, contribute to the sexually dimorphic susceptibility of fruit flies to *P. alcalifaciens* infection. Specifically, male flies exhibit elevated expression of a key iron transporter, *Tsf1*, compared to females. This resulted in a lower amount of iron in the hemolymph of males. Given that iron is an essential element for bacterial growth, lower iron content in males leads to reduced pathogen growth and significantly longer survival of male flies. Importantly, while *Tsf1* expression was consistently higher in males before and during infection, sex differences in iron content were significant only under basal conditions. Given that the initial phase of infection predicts the

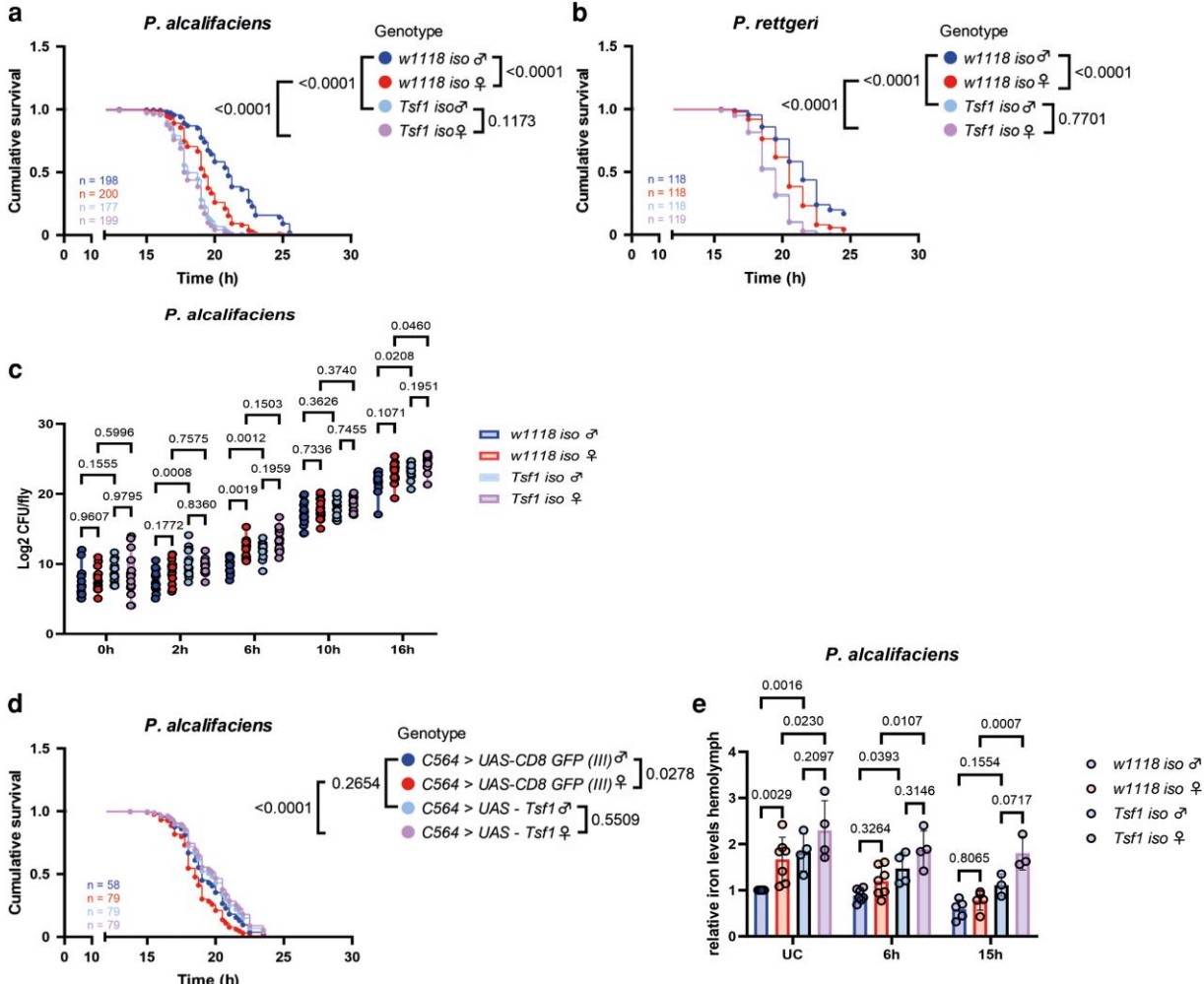

**Fig. 3.** Differences in *Tsf1* expression cause the sexual dimorphism in survival and iron content. a to b) Cumulative survival of adult flies infected with *P. alcalifaciens* (a) and *P. rettgeri* (b), respectively. *n* indicates the total number of flies used in the experiments per genotype. c) *P. alcalifaciens* load at 0, 2, 6, 10, and 16 h postinfection. For CFU counts, 1 dot represents the pathogen load of 1 infected fly. Results are shown as mean and SD of 3 independent experiments. Data were analyzed using two-way ANOVA and Tukey's multiple comparisons test. d) Cumulative survival of flies overexpressing *Tsf1* (*C564>UAS-Tsf1*) or GFP as a negative control (*C564>UAS-CD8-GFP*) in the fat body. *n* indicates the total number of flies used in the experiments per genotype. e) Relative iron levels in the hemolymph of unchallenged (UC) flies or after infection with *P. alcalifaciens* after 6 and 15 h. The iron content was measured using the ferrozine assay. One dot represents the iron levels of a pool of 75+ flies. Data shown as relative values compared to iron levels in unchallenged *w1118 iso* male flies. The mean and SD of at least 3 independent experiments are shown. Data were analyzed using two-way ANOVA and Tukey's multiple comparisons test.

probability of survival in *Drosophila* (Duneau et al. 2017), basal differences between males and females in the amount of iron that is available to pathogens are likely to be sufficient to determine the sex-biased infection outcome. Indeed, we detected higher pathogen loads in female flies at early stages of infection, which likely stems from an increased availability of iron.

Further support for the link between iron availability and susceptibility to *P. alcalifaciens* infection comes from FAC feeding and *Tsf1* overexpression experiments. The higher iron load in the flies fed with FAC significantly increased the susceptibility of males but not females to *P. alcalifaciens* infection, while decreasing iron in the hemolymph by *Tsf1* overexpression increased female but not male survival after infection. Such sex-biased effects of these treatments could be attributed to the pre-existing sex differences in iron content. Since the basal iron load in females is higher compared to males, further increase by FAC feeding has no consequences for their survival. While with *Tsf1* overexpression, the situation is opposite—further decrease of an already low iron content is not relevant for male survival. Alternatively, basal low iron

levels in males might be achieved by the higher expression or activity of additional iron regulators, such as ferritin or Malvolio, making *Tsf1* overexpression less impactful. Together, FAC feeding and *Tsf1* overexpression experiments support a role of sexual dimorphism in iron metabolism and in sex-biased susceptibility to infection.

The sex differences in iron sequestration that we identified could potentially explain other sexually dimorphic traits in *Drosophila*. Since iron is a redox-active metal, its excess can promote the generation of ROS. Hence, elevated iron levels in the hemolymph of female flies might be the reason for their increased ROS levels and susceptibility to oxidative stress (Albrecht et al. 2011; Rubinić et al. 2025). On the other hand, increased ROS levels might provide protection to female flies and explain their increased resistance against certain pathogens, like *S. aureus*, which is believed to be susceptible to immune-induced ROS (Dudzic et al. 2019). Since certain pathogens, like the microsporidian *Nosema ceranae*, and endosymbionts, such as *Spiroplasma*, utilize iron complexed with transferrin as an iron source within the host

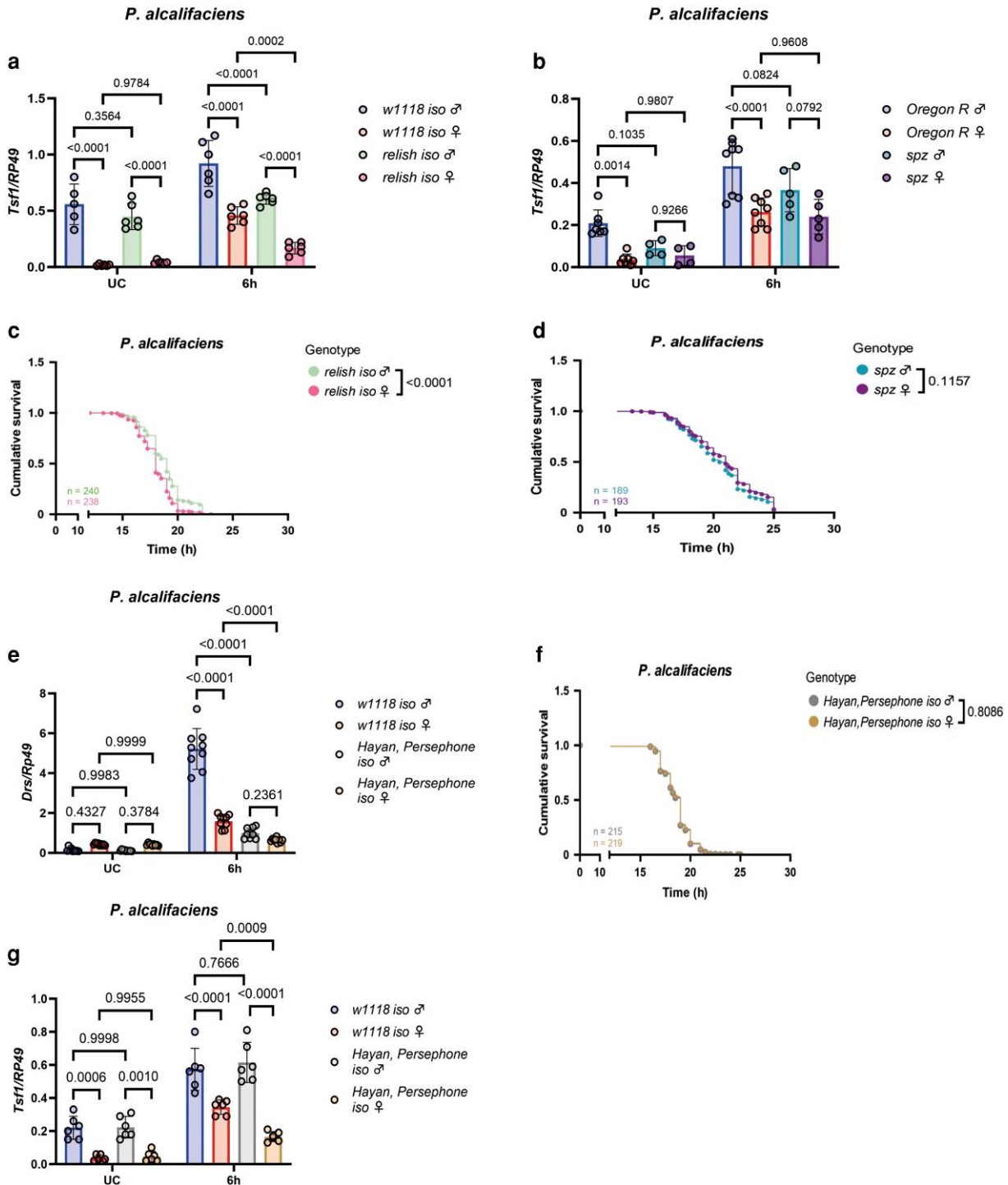

**Fig. 4.** Differences in Toll pathway activation induce the sexually dimorphic *Tsf1* expression and fly survival. a to b) RT-qPCR measuring *Tsf1* expression in IMD- (*relish iso*) (a) and Toll (*spz*) (b) pathway mutants. Data show unchallenged (UC) or flies infected with *P. alcalifaciens* 6 h postinfection, respectively. One dot represents expression levels from 8 to 10 pooled flies. The mean and SD of at least 2 independent experiments are shown. Data were analyzed using two-way ANOVA and Tukey's multiple comparisons test. c to d) Cumulative survival of immune mutants *relish iso* (IMD pathway) (c) and *spz* (Toll pathway) (d). *n* indicates the total number of flies used in the experiments per genotype. e) RT-qPCR measuring *Drs* expression in unchallenged (UC) or flies infected with *P. alcalifaciens* 6 h postinfection, respectively. One dot represents expression levels from 10 pooled flies. The mean and SD of 3 independent experiments are shown. Data were analyzed using two-way ANOVA and Tukey's multiple comparisons test. f) Cumulative survival of *Hayan, Persephone iso* flies. *n* indicates the total number of flies used in the experiments per genotype. g) RT-qPCR measuring *Tsf1* expression in unchallenged (UC) or flies infected with *P. alcalifaciens* 6 h postinfection, respectively. One dot represents expression levels from 10 pooled flies. The mean and SD of 3 independent experiments are shown. Data were analyzed using two-way ANOVA and Tukey's multiple comparisons test.

environment (Marra et al. 2021; Rodríguez-García et al. 2021), increased availability of Tsf1 in male flies might render them more susceptible to such pathogens.

We found that the Toll pathway controls basal differences between males and females in *Tsf1* expression. Given a previously demonstrated elevated activity of the Toll pathway in male flies

(Duneau et al. 2017), elevated *Tsf1* expression and iron sequestration in males are likely a consequence of higher Toll pathway activity. Hence, we identified Tsf1 as one of the Toll pathway-regulated immune effectors involved in immune dimorphism, further deepening our understanding of the mechanisms underlying sex-biased susceptibility to infection. Notably, while our results are consistent with the previously demonstrated role of pathogen sensing by Psh in the sexually dimorphic activation of the Toll pathway (Duneau et al. 2017), Psh, in contrast to spz, was not required for basal sex differences in *Tsf1* expression. This phenotype is consistent with a specific role of Psh in Toll activation during an immune response. Alternatively, *Psh-Hayan* could contribute to sexual dimorphism by regulating melanization response (Dudzic et al. 2019; Shan et al. 2023). However, this is an unlikely scenario, as melanization-deficient *PPO1⁴,2⁴* mutant still exhibits sex differences in survival, suggesting that melanization does not play a role in the dimorphism. The fact that *Psh-Hayan* mutants retain sex differences in *Tsf1* expression but lose sex dimorphism in survival and Toll activation suggests that besides Tsf1, additional Toll-regulated immune effectors contribute to sex differences in susceptibility to infection. However, in the case of *P. alcalifaciens*, Tsf1 is a key mediator of sexual dimorphism as sex differences in survival were abrogated in *Tsf1*-deficient flies. Notably, while the differences in iron levels in the *Tsf1* mutant males and females were not statistically significant, there was still a trend toward higher iron amount in *Tsf1* females. This indicates that, in addition to Tsf1, there are likely to be additional iron transporters, which remain to be identified, that might mediate sex differences in the hemolymph iron content.

The important remaining question is what determines sexual dimorphism in the Toll pathway. Given the established interactions between immunity and hormones in *Drosophila* (Regan et al. 2013; Schwenke et al. 2016), hormonal differences between females and males are likely contributors to differential Toll pathway activation. For instance, females produce juvenile hormone (JH) which is crucial for reproduction. JH suppresses the immune response with a stronger influence on the Toll pathway than on the Imd pathway (Flatt et al. 2008; Schwenke and Lazzaro 2017). Therefore, the female-biased production of JH has a cost on their immune system, leading to sexual dimorphism in the Toll pathway activity and response to certain infections (Duneau et al. 2017). Additionally, JH can inhibit the immune response indirectly by suppressing 20-Hydroxyecdysone (20E)—a known potentiator of immunity (Dimarcq et al. 1997; Flatt et al. 2008; Schwenke et al. 2016). Higher titers of JH in females than in males could lead to a stronger inhibition of 20E and consequent reduced activation of the Toll pathway. Additionally, sexually dimorphic constitutive expression of Toll signaling genes could also contribute to sexual dimorphism in activation of the Toll pathway. Indeed, genes that encode extracellular components of the Toll pathway with a specific role in immune response, including pattern recognition receptors and serine proteases, show higher constitutive expression in males than females. Females express stronger Toll pathway genes required for both embryonic development and immune response (intracellular part of the Toll pathway) (Duneau et al. 2017). Therefore, dimorphism in constitutive expression of Toll pathway genes likely determines subsequent differences in infection outcome between sexes. Given an established immunomodulatory role of microbiota (Tafesh-Edwards and Eleftherianos 2023), another possibility could be that reported differences in microbiota composition between males and females result in sex-biased Toll pathway activity (Leech et al. 2021; Yu and Iatsenko 2025).

Given that we still observed sex differences (at the edge of statistical significance) in *Tsf1* expression under infection conditions in *spz* mutants, there are likely other pathways that control *Tsf1* induction in a sex-biased manner. One of these pathways could be the Janus kinase (JAK)-signal transducer and activator of transcription (STAT) pathway, which regulates the expression of immune effectors. The fact that some of these effectors, such as *Turandot C (TotC)*, *TotA*, and *TotX*, were induced by infection in male but not female flies (Duneau et al. 2017; Khan and Graze 2024) raises the possibility that infection-induced sex differences in *Tsf1* expression stem from the sexually dimorphic activation of the Jak-STAT pathway. Sex differences in *Tsf1* expression could also be driven by the sex bias in the activity of insulin/insulin-like growth factor signaling (IIS) and its downstream transcription factor Forkhead Box O (FOXO)—a known regulator of AMP expression (Becker et al. 2010). Higher activity of IIS in females (Biswas et al. 2025) and the subsequent lower activation of FOXO could lead to reduced expression of immune effectors, including *Tsf1* in female flies.

Sex bias in iron sequestration by Tsf1 likely contributes to sexually dimorphic susceptibility not only to the here tested *Providencia* sp. but to a broader range of pathogens. For instance, we would anticipate this mechanism to be at play during infections with *Pseudomonas* sp. and *Mucorales* fungi, given a previously reported prominent role of Tsf1-mediated iron sequestration in the defense against these pathogens (Iatsenko et al. 2020). Indeed, male-biased iron sequestration could be one of the mechanisms contributing to increased resistance of males to *P. entomophila*—a pathogen controlled by Tsf1-mediated iron sequestration (Rubinić et al. 2025). However, the contribution of Tsf1-mediated iron sequestration to the sexual dimorphism in infection outcome will depend on how important iron sequestration is in the defense against a particular pathogen relative to the other immune arms, some of which might also be sexually dimorphic. Dissection of the potential sexual dimorphism in the other *Drosophila* immune defense reactions will provide valuable insights into the biological processes that contribute to sex-specific infection outcome.

## Data availability

The authors affirm that all data necessary for confirming the conclusions of this article are represented fully within the article and its tables and figures.

Supplemental material available at GENETICS online.

## Acknowledgments

We are grateful to the Bloomington *Drosophila* Stock Center (NIH P40OD018537) for fly stocks. We thank Dagmar Frahm for technical assistance.

## Funding

This work was supported by the Max Planck Society and Deutsche Forschungsgemeinschaft (IA 81/2-1). I.I. also acknowledges the funding from the Deutsche Forschungsgemeinschaft (IA81/3-1) and Boehringer Ingelheim Foundation.

## Conflicts of interest

None declared.

## Author contributions

Alexandra Hrdina (Investigation, Formal analysis, Data curation, Methodology, Visualization, Writing—original draft, Writing—review & editing), Igor Iatsenko (Conceptualization, Funding acquisition, Project administration, Supervision, Resources, Writing—original draft preparation, Writing—review & editing)

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

*Editor: H. Jafar-Nejad*