## [Peer Review File · Genetics]

Sex Bias in Iron Sequestration by Transferrin 1 Modulates Sexually-Dimorphic Infection Outcomes in *Drosophila melanogaster*

Alexandra Hrdina and Igor Iatsenko

NOTE: The reviews and decision letters are unedited and appear as submitted by the reviewers.

In extremely rare instances and as determined by a Senior Editor or the EIC, portions of a review may be redacted. If a review is signed, the reviewer has agreed to no longer remain anonymous.

The review history appears in chronological order.

Review Timeline:

Submission Date:	2025-08-07
Editorial Decision:	2025-09-02
Resubmission Received:	2026-01-28
Editorial Decision:	2026-02-16
Revision Received:	2026-02-24
Accepted:	2026-02-24

September 2, 2025

GENETICS-2025-308469

Sex Bias in Iron Sequestration by Transferrin 1 Modulates Sexually-Dimorphic Infection Outcomes in *Drosophila melanogaster*

Dear Dr. Iatsenko:

Three experts in the field have reviewed your manuscript. The reviewers have found the manuscript to be of broad interest and potentially important. However, the reviewers also had a relatively large number of major and minor concerns about the manuscript, related to various issues including data interpretation, statistical analysis, and controls. Therefore, unfortunately, your manuscript is not currently acceptable for publication in GENETICS. If you are willing to address the reviewers' comments in a substantially revised manuscript, we will be happy to reconsider it. Please note that based on the reviews, a revised version will require a significant number of additional experiments. You can read their reviews at the end of this email. If you decide to submit a revised manuscript, please let the editorial office know approximately how long you expect to need for revisions. We understand that these revisions might take 4-6 months.

Upon resubmission, please include:

1. A clean version of your manuscript;
2. A marked version of your manuscript in which you highlight significant revisions carried out in response to the major points raised by the editor/reviewers (track changes is acceptable if preferred);
3. A detailed response to the editor's/reviewers' feedback and to the concerns listed above. Please reference line numbers in this response to aid the editor and reviewers.

Your paper will likely be sent back out for review.

Additionally, please ensure that your resubmission is formatted for GENETICS
<https://academic.oup.com/genetics/pages/general-instructions>

Follow this link to submit the revised manuscript: Link Not Available

Sincerely,

Hamed Jafar-Nejad
Associate Editor
GENETICS

Approved by:
David Greenstein
Senior Editor
GENETICS

Reviewer #1 :

Summary

This manuscript, "Sex Bias in Iron Sequestration by Transferrin 1 Modulates Sexually-Dimorphic Infection Outcomes in *Drosophila melanogaster*," addresses the molecular basis of sex differences in infection outcomes in *Drosophila melanogaster*, focusing on iron sequestration mediated by Transferrin 1 (Tsf1). The authors link elevated Tsf1 expression in males to lower hemolymph iron levels and greater resistance to *Providencia* infection, with the Toll pathway implicated in sex-biased regulation. These findings contribute to the emerging field of sex-specific immunity and nutritional immunity. The work is timely and of interest to readers; however several aspects require clarification, additional analysis, or refinement to strengthen the rigor and generalizability of the conclusion.

Major Concerns:

1. The FAC feeding experiment suggests that males experience a marked decrease in survival (Fig. 2d) while females appear unaffected. This raises the question of whether iron uptake/storage is sex-biased? Males may absorb or mobilize iron differently, leading to greater iron overload after FAC feeding.

Alternatively, oxidative stress may play a role, as iron overload can enhance ROS production via Fenton chemistry. If males are more sensitive to oxidative stress, that could explain their sharp survival decline. Measuring hemolymph iron levels after FAC feeding would help determine whether iron loading is equivalent between sexes. For example, females might resist dietary iron

accumulation through ferritin storage or excretion, explaining why their survival is unchanged.

2. The observation that Tsf1 mutant females survive similarly to males (Fig. 3c) despite higher pathogen loads (Fig. 3d) is intriguing and potentially important for the tolerance literature. However, tolerance here is inferred from survival-pathogen load patterns. A more formal analysis (e.g., survival vs. pathogen load relationships across a gradient of infection doses) would be needed to support this claim, or the interpretation should be framed more cautiously as a hypothesis.

3. The study only includes CFU measurements at 6 h and 16 h post-infection (Fig. 1d, 3c). These timepoints are too coarse to capture the dynamics of bacterial clearance versus proliferation.

Without additional temporal data, the authors cannot distinguish whether males are better at initial pathogen killing or merely slow proliferation later. This weakens the mechanistic interpretation of resistance versus tolerance.

3. While the data indicate Toll pathway involvement in sex-biased Tsf1 expression (Fig. 4a-d), the underlying reason for elevated Toll activity in males remains speculative. The discussion would benefit from a more critical evaluation of potential upstream regulators (e.g., hormonal modulation, microbiota differences, basal immune activation), and whether the observed dimorphism is specific to the Toll-Tsf1 axis or reflects broader Toll pathway bias.

4. In addition, the fact that spz mutants retain infection-induced Tsf1 expression (Fig. 4b), suggests that other pathways, e.g., JAK-STAT or Insulin signaling, might contribute to the difference.

4. The work primarily uses *P. alcalifaciens* (and *P. rettgeri* to a lesser extent) as infection models. Since Tsf1 has been implicated in defense against multiple pathogens, the authors should comment on whether similar sex-biased effects are expected for Gram-positive bacteria, fungi, or viruses. Even a short discussion of the predicted breadth of this mechanism would strengthen the impact.

5. The virgin versus mated fly data suggest reproduction is not the primary driver of female susceptibility (Fig.1). However, the potential contribution of juvenile hormone or other reproductive hormones to Toll pathway activity is mentioned only briefly in the discussion. A more detailed exploration of hormonal regulation, even speculative, would strengthen the mechanistic framework.

6. Most experiments were performed in w1118 iso. Although Oregon R was included for survival analysis, it would be reassuring to show iron levels or Tsf1 expression differences in a second genetic background. This would ensure that the phenotype is not background-specific.

7. It is not always clear whether survival, iron, or qPCR replicates derive from the same cohort or from independent generations. Given the variability in microbial load and nutritional state across fly batches, additional detail on the replication strategy is essential.

Minor comments:

1. Figures and text should specify the statistical tests used for each comparison, and whether corrections for multiple comparisons were applied. While the methods mention Cox proportional hazard models and ANOVA, the figure legends occasionally omit these details.

2. The iron measurements are normalized to total protein. The authors should confirm that there are no systematic sex differences in protein content that could affect normalization.

3. In Materials and Methods, please clarify whether experimental flies were age-matched between sexes, and whether males and females were housed separately prior to experiments to control for social/environmental effects.

4. In Figures 2 and 3, some dots representing biological replicates are difficult to distinguish due to overlap. Consider using jitter or transparency for better visibility.

5. In the methods, specify whether each qPCR replicate represents pooled flies from independent cohorts or repeated measurements of the same RNA sample.

6. Gene names (e.g., Tsf1, spz, Relish) should be italicized, while protein names should not. Please ensure consistency throughout.

7. For spz and Relish mutants, specify whether backcrossing to w1118 iso was performed to minimize background effects.

8. Exact sample sizes are not consistently provided in figure legends. For survival curves, "n" is reported as total flies, but it is not clear how many vials or replicates were used.

9. Only Rp49 was used for normalization in qPCR. A single housekeeping gene may not be stable across sexes or infection conditions. Validation of multiple reference genes would be preferable.

10. Survival of FAC-fed uninfected flies should be included as a control to ensure FAC is not toxic or affecting lifespan independently of infection.

Reviewer #2 :

The present study investigates the role of iron sequestration by Transferrin 1 in mediating sex differences to infection. This work follows from a previous study where Tsf1 was shown to interact with *P. alcalifaciens*. The study shows females succumb to *P. alcalifaciens* to a greater extent, have higher iron loads consistent with Tsf1 expression levels, and the susceptibility to *P. alcalifaciens* is lost in Tsf1 mutants, even if sexually dimorphic expression differences are not. This provides a solid set of initial findings. However, I feel more could be done to provide a conceptual advance, and to shore up the rigour of experiments in this manuscript.

Major comments

1. It's not clear if the loss of sexual dimorphism in Tsf1 mutants has more to do with a maximum growth rate of the bacteria in a relatively resistance-deficient fly strain. Indeed, if one squints, males still survive slightly longer than females in Fig 3A ($P = 0.11$). The expression of Tsf1 is sexually dimorphic even in Rel and spz mutant backgrounds, despite these backgrounds differing in their extent of sexual dimorphism in survival. Overall, this suggests that Tsf1 mutation causes susceptibility so severe that sexual dimorphism is muted relative to the effect of immune deficiency. But this is not the same as Tsf1 being causative of the sexual dimorphism. For instance, Relish mutants still retain a sexual dimorphism (Fig 4C), but have impaired Tsf1 induction in both sexes. Fig4B shows Tsf1 is actually impaired at baseline and induced state in spz relative to Rel, given the Tsf1/RP49 ratio. Showing the sexual dimorphism of Fig3A over a range of doses, or perhaps temperatures to slow the pathogen growth and improve resolution, would confirm if Tsf1 mutants retain a sexual dimorphism or not.

2. Following from #1 Attributing the spz difference to Tsf1 is speculative and not demonstrative, particularly given a previous study reported this loss of sexual dimorphism in spz mutants was paralleled by Persephone, relying on microbial protease sensing and not iron sequestration (Duneau et al., 2017 BMC Biology). Investigations of Psh and Tsf1 alone or combined would strengthen the conclusions here, test the reproducibility of the Psh finding (which has not been repeated since 2017, if I'm correct), and provide a more valuable advance for the field.

3. Fig3C: The higher pathogen loads here are taken as an idea that females have higher tolerance to *Providencia* than males (Abstract line 37). But females are larger period, have more hemolymph for bacteria to grow in, and also more iron to promote a faster growth. So can this not simply reflect a higher pathogen load needed to kill females because they're bigger? An argument of tolerance would require showing that the normalized bacterial load per body size (ex: wet mass, dry mass) is somehow different in males vs females. It would be reasonable to say that females increase pathogen virulence by having more iron, but I'm not sure this is really paralleled as "lower tolerance" in the way those terms are used. If nothing more, greater nuance is needed in dissecting the underlying factors behind this result, and I would not simplify it to "lower tolerance," or at least, not without qualifying and justifying what is meant by this. Ideally, some additional body size data could be incorporated into the analysis, which would be relatively quick to do, and improve the ability of the authors to speak to virulence, resistance, and tolerance.

4. Line 55, 300: do the authors find females have higher ROS in their settings? In general, an exploration of the melanization reaction and Tsf1 in these sex differences would improve the study. Presently, the argument is made that Tsf1 deploys its effect through nutritional immunity, but it could easily be that differences in free iron lead to differences in the potency of the ROS response mediated by melanization, which does not have to do with nutritional immunity. This would provide a more valuable advance for the field.

Minor comments

I appreciate the care to put sample sizes in each figure, which aids the reader to assess the weight of the evidence.

Fig1D, Line 200, Fig3C: are the p-values above the boxplots accurate? It's not clear to me how the 6h time point in Fig1D can effectively be $P = 1.00$ with the spread of data points shown. The same comment stands for 6h in Fig3C comparing males, where the cluster of data points hardly overlaps at all, yet somehow the 16h male-female comparisons are significant. Have the authors analyzed raw values, or log-transformed data? I would recommend analyzing Log2 log-transformed data. i.e. transform the data before running the analysis. Log2 as a transformation because bacteria divide in a 2-fold manner, not a 10-fold manner, so this treatment better reflects a meaningful difference in the log-transformed data in terms of replication cycles achieved within the host. Without this data transformation, differences of thousands of bacteria within a genotype within timepoint (such as the min/max ranges seen in Fig3C), create an artificial skew to the standard deviation and reducing all statistical power. This would explain the binary trend of "significant" or " $P > 0.9999$ " the authors show in boxplots.

Fig2A and FigS1: it is important the authors demonstrate this Tsf1 effect does not rely on genetic background. The authors should support the FlyAtlas2 data with survival using Canton S. The authors should also check Tsf1 expression of OR-R, as they have for the iso line. These quick experiments would strengthen the rigour and conclusions of the manuscript.

Fig3E, Line 247: It looks like differences in iron load among sexes are present at all time points in Fig3E, even if reduced in magnitude. Strangely, there is not an equal number of data points in each bar, which might be obfuscating some of the differences. For instance, the trend at 15h is quite strong, with no overlap at all, but only marginally significant ($P = .07$) because only 3 data points are present for Tsf1 in each sex. Moreover, concerns on how the statistics were done for these sorts of plots remain, so I would encourage the authors to 1st) look at the trends and consider the language used in the text, and whether this is justified, independent of what the P-values say. After all one can somewhat modify P-values with justifiable analytical choices (ex: two-way ANOVA vs. glm, or repeated T-tests with FDR correction vs. bonferroni correction, etc...). But the observed consistent trends should not be dismissed simply because within-timepoint comparisons were just off of 'significance'. 2nd) reevaluate the statistical analyses and ensure they are robust. Indeed, the choice of whether to run a whole-data two-way ANOVA with genotype-by-sex, and time point as covariates would likely reveal the interaction term that Tsf1 females have higher iron loads, even if individual time points might fail to reach significance at the current level of sampling within timepoints.

Line 215: significantly? Please provide the test type, test statistic, and P-value here to confirm what is being discussed. Is this an all-timepoints analysis with timepoint and genotype as covariates? Two-way ANOVA? Glim?

Line 90: clarify grammar

Fig3D: are the sample sizes accurate? If so, this experiment must be repeated with 2+ additional replicates as all other experiments have been.

Line 309: the Toll pathway did not control basal- or infection-based differences in Tsf1 expression, right? Tsf1 remained inducible in spz mutants, and sex differences were consistent, even if lower compared to wild-type.

Line 330-332: no discussion of Persephone.

The authors have not made their data available for independent review. The data are available in the manuscript and figures, i.e. not available. Even if the authors just wished to upload Prism files to a FigShare, that would at least allow an independent review to determine the analysis results. For instance, I could better understand what has gone off in the authors' statistical analyses if the data were available.

Reviewer #3 :

In this manuscript, GENETICS-2025-308469, the authors investigated the mechanistic basis of sexual dimorphism in infection outcomes using *Drosophila melanogaster*. The work is focused on nutritional immunity, specifically iron sequestration mediated by Transferrin 1 (Tsf1). The authors showed that males exhibit higher basal and infection-induced expression of Tsf1, resulting in lower iron levels in the hemolymph and improved survival following *Providencia* infection. Importantly, Tsf1 mutants abolish sex differences in survival but retain differences in bacterial load, leading the authors to propose that Tsf1 contributes not only to resistance but also to tolerance. Finally, they demonstrate that sex-biased Tsf1 expression is mediated by the Toll pathway. This is a well-executed study that offers a novel mechanistic insight into sexual dimorphism in immunity via iron sequestration. The work provides an informative contribution to the fields of immunity, host-pathogen interactions, and sexual dimorphism.

The manuscript is clearly written, and the experimental design takes strong advantage of the *Drosophila* system. While some interpretations (particularly regarding sequestering and tolerance) could be better substantiated, the manuscript is of broad interest to geneticists and immunologists. There are the following major and minor points.

Major Points

1. Based on the data presented in Figure 2, the connection between Tsf1 expression (Fig. 2b) and hemolymph iron levels (Fig. 2c) needs clearer interpretation. Males show higher Tsf1 expression both basally and after infection, which should translate into lower hemolymph iron level. This matches the basal data, but after infection iron levels in males and females are comparable. Although, the authors do not seem to compare hemolymph iron levels in male and females at the different time points (0hr, 1.5hr,.....to 16h post-infection), the trend (Fig 2c) shows that females sequester strongly. In other words, the expression and iron measurements do not fully align. This raises the possibility that baseline iron differences, rather than infection-induced sequestration, are the main driver of the survival phenotype. Since, male have high Tsf1 levels and less free iron, so less support for bacteria and more survival. Or there could be additional regulators besides Tsf1 shaping hemolymph iron during infection. The discussion and interpretation should be explicitly revised to reflect this.

2. There is not enough data to support the interpretation of tolerance in Tsf1 mutants. In Tsf1 mutants, sex differences in survival disappear, but females still carry a higher bacterial burden (Fig. 3c). Based on these data, survival and bacterial loads are not aligned; therefore, the idea of tolerance is reasonable but not supported by any data which can include demonstration of

reduced detrimental readouts e.g., ROS; oxidative damage. Please consider either including data or revising to tone down the conclusion.

3. The authors showed that overexpressing Tsf1 improves survival in females (potentially by increasing Tsf1 level and subsequently hemolymph iron level) but not in males. The authors explained this as a ceiling effect in males. There is an experimental caveat that the hemolymph iron levels were not shown. Please consider adding data to support the ceiling effect or revise the interpretation and discussion for other explanations, the contribution of other iron regulators such as ferritin or Malvolio.

4. The authors showed that Toll pathway activity explains why males express more Tsf1 than females. But the cause of this sex difference in Toll signaling isn't addressed. They briefly mentioned juvenile hormone (JH). Given the major focus of this manuscript on the underlying mechanism of sex dimorphism, it would be important to have a substantial discussion covering other possible mechanisms to fill this gap.

5. The authors demonstrated male-biased Tsf1 effects in two *Providencia* species. That's solid, but it doesn't prove the mechanism is universal. For the broad scope of this manuscript, it would be interesting to know if it is a general principle of nutritional immunity in flies or if it is a pathogen-specific effect that happens to show up in *Providencia*. Please consider adding a discussion segment explicitly addressing whether the same sex-specific effects are expected in other bacterial or microbial infections.

Minor Points

1. In Abstract, please consider introducing Tsf1 as transferrin-1. In the current version, the Tsf1 appears in a manner that assumes prior knowledge and may confuse general readers.
2. Please revise for typographical issues. For example, line 152- "the the SYBR Select Master Mix". Please remove duplicate "the". Also in line 157: "on two 10 μ m filter". It should be "filters."
3. Please, check Figure legend 3 for italicized gene and genotype names.
4. In Figure legends, please define p-value.

Associate Editor Comments:

Point by point responses to the referees comments on GENETICS-2025-308469 manuscript entitled "Sex Bias in Iron Sequestration by Transferrin 1 Modulates Sexually-Dimorphic Infection Outcomes in *Drosophila melanogaster*"

Reviewer #1 :

Summary

This manuscript, "Sex Bias in Iron Sequestration by Transferrin 1 Modulates Sexually-Dimorphic Infection Outcomes in *Drosophila melanogaster*," addresses the molecular basis of sex differences in infection outcomes in *Drosophila melanogaster*, focusing on iron sequestration mediated by Transferrin 1 (Tsf1). The authors link elevated Tsf1 expression in males to lower hemolymph iron levels and greater resistance to *Providencia* infection, with the Toll pathway implicated in sex-biased regulation. These findings contribute to the emerging field of sex-specific immunity and nutritional immunity. The work is timely and of interest to readers; however several aspects require clarification, additional analysis, or refinement to strengthen the rigor and generalizability of the conclusion.

Response. We thank the reviewer for such a comprehensive summary of our work and constructive suggestions.

Major Concerns:

1. The FAC feeding experiment suggests that males experience a marked decrease in survival (Fig. 2d) while females appear unaffected. This raises the question of whether iron uptake/storage is sex-biased? Males may absorb or mobilize iron differently, leading to greater iron overload after FAC feeding.

Alternatively, oxidative stress may play a role, as iron overload can enhance ROS production via Fenton chemistry. If males are more sensitive to oxidative stress, that could explain their sharp survival decline. Measuring hemolymph iron levels after FAC feeding would help determine whether iron loading is equivalent between sexes. For example, females might resist dietary iron accumulation through ferritin storage or excretion, explaining why their survival is unchanged.

Response. Regarding sensitivity to oxidative stress, our work <https://www.biorxiv.org/content/10.1101/2025.05.22.655590v1.abstract> and this paper <https://www.sciencedirect.com/science/article/pii/S1550413111004190> showed that males are more resistant to oxidative stress. Following the recommendation, we measured hemolymph iron in males vs females after FAC feeding. We found that FAC feeding led to a significant increase in hemolymph iron level in both sexes. Females accumulated even more iron than males, while males after FAC feeding reached iron levels of sucrose-fed females (new Figure 2d, Lines 230-232). Hence, females do not simply resist iron accumulation. It is likely that once iron levels reach a certain threshold, additional increases are not relevant to the infection outcome, explaining why higher iron amount after FAC feeding does not further increase female susceptibility to infection.

2. The observation that Tsf1 mutant females survive similarly to males (Fig. 3c) despite higher pathogen loads (Fig. 3d) is intriguing and potentially important for the tolerance literature. However, tolerance here is inferred from survival-pathogen load patterns. A more formal analysis (e.g., survival vs. pathogen load relationships across a gradient of infection doses) would be needed to support this claim, or the interpretation should be framed more cautiously as a hypothesis.

R. The same point was also raised by the other 2 reviewers. We agree that we have insufficient data to fully support the role of Tsf1 in tolerance. Additionally, inclusion of additional timepoints and reanalysis of log₂-transformed data (as suggested by the reviewer 2) did not reveal statistically significant differences in pathogen load between Tsf1 male and female flies. Therefore, we removed the claim about tolerance from the manuscript.

3. The study only includes CFU measurements at 6 h and 16 h post-infection (Fig. 1d, 3c). These timepoints are too coarse to capture the dynamics of bacterial clearance versus proliferation. Without additional temporal data, the authors cannot distinguish whether males are better at initial pathogen killing or merely slow proliferation later. This weakens the mechanistic interpretation of resistance versus tolerance.

R. We performed CFU measurements at additional timepoints, as recommended (Updated figures 1d, 3c). The obtained results support the hypothesis of better pathogen control by males at initial stages of infection. Please note that *P. alcalifaciens* is a fast killer, therefore differences in phenotypes with high variability, like CFU, might be challenging to uncover.

3. While the data indicate Toll pathway involvement in sex-biased Tsf1 expression (Fig. 4a-d), the underlying reason for elevated Toll activity in males remains speculative. The discussion would benefit from a more critical evaluation of potential upstream regulators (e.g., hormonal modulation, microbiota differences, basal immune activation), and whether the observed dimorphism is specific to the Toll-Tsf1 axis or reflects broader Toll pathway bias.

R. We extended the discussion and covered the contribution of additional factors to the sexual dimorphism in the Toll pathway activation, including additional hormones, microbiota differences, and differences in basal level of Toll pathway activity. Lines 370-385.

4. In addition, the fact that spz mutants retain infection-induced Tsf1 expression (Fig. 4b), suggests that other pathways, e.g., JAK-STAT or Insulin signaling, might contribute to the difference.

R. We agree and discussed the other pathways that can potentially control sex differences in infection-induced Tsf1 expression (L 386-398).

4. The work primarily uses *P. alcalifaciens* (and *P. rettgeri* to a lesser extent) as infection models. Since Tsf1 has been implicated in defense against multiple pathogens, the authors should comment on whether similar sex-biased effects are expected for Gram-positive bacteria, fungi, or viruses. Even a short discussion of the predicted breadth of this mechanism would strengthen the impact.

R. A similar recommendation was also made by the reviewer 3. We covered this point in the Discussion, L 399-411.

5. The virgin versus mated fly data suggest reproduction is not the primary driver of female susceptibility (Fig.1). However, the potential contribution of juvenile hormone or other reproductive hormones to Toll pathway activity is mentioned only briefly in the discussion. A more detailed exploration of hormonal regulation, even speculative, would strengthen the mechanistic framework.

R. We extended the Discussion on the role of hormones (L 370-374).

6. Most experiments were performed in w1118 iso. Although Oregon R was included for survival analysis, it would be reassuring to show iron levels or Tsf1 expression differences in a second genetic background. This would ensure that the phenotype is not background-specific.

R. We have measured Tsf1 expression and iron levels in Oregon R background and observed a similar to w1118 iso phenotype. See new Figure S2a-b.

7. It is not always clear whether survival, iron, or qPCR replicates derive from the same cohort or from independent generations. Given the variability in microbial load and nutritional state across fly batches, additional detail on the replication strategy is essential.

R. We added a description to the materials and methods (L 136-137, L159-160, L177-179).

Minor comments:

1. Figures and text should specify the statistical tests used for each comparison, and whether corrections for multiple comparisons were applied. While the methods mention Cox proportional hazard models and ANOVA, the figure legends occasionally omit these details.

R. We added these details to the legends.

2. The iron measurements are normalized to total protein. The authors should confirm that there are no systematic sex differences in protein content that could affect normalization.

R. We apologize for imprecise description of the procedure. We used equal protein loading for each sample (120 µg of total protein) rather than normalization to total protein. This was corrected in the methods section (L169-170). Such an approach is commonly used in the field (Xiao et al, Cell Reports 2019, Weber et al., Insect Bio Mol Biol. 2022).

3. In Materials and Methods, please clarify whether experimental flies were age-matched between sexes, and whether males and females were housed separately prior to experiments to control for social/environmental effects.

R. These details were added to the Method Section under *Drosophila* stocks and rearing (L120-124).

4. In Figures 2 and 3, some dots representing biological replicates are difficult to distinguish due to overlap. Consider using jitter or transparency for better visibility.

R. Modified.

5. In the methods, specify whether each qPCR replicate represents pooled flies from independent cohorts or repeated measurements of the same RNA sample.

R. Specified in the materials and methods section (L159-160).

6. Gene names (e.g., Tsf1, spz, Relish) should be italicized, while protein names should not. Please ensure consistency throughout.

R. Done.

7. For *spz* and *Relish* mutants, specify whether backcrossing to *w1118* iso was performed to minimize background effects.

R. *Relish* mutant was in *w1118* iso background but *spz* mutant was Oregon R background. We performed survivals also with *spz* mutant in *w1118* iso background (see graph below) and observed a similar loss of sexual dimorphism as with Oregon R background.

8. Exact sample sizes are not consistently provided in figure legends. For survival curves, "n" is reported as total flies, but it is not clear how many vials or replicates were used.

R. These details were added to the Method Section.

9. Only *Rp49* was used for normalization in qPCR. A single housekeeping gene may not be stable across sexes or infection conditions. Validation of multiple reference genes would be preferable.

R. *RP49* is a well-established reference gene commonly used in *Drosophila* studies, including those on sex differences in immunity. Since with our qPCR we could reproduce previously reported RNA-seq results on sex differences in basal (FlyAtlas) and infection-induced (Duneau et al, BMC Biology 2017) sex differences in *Tsf1* expression, we believe that *RP49* is an appropriate reference control.

10. Survival of FAC-fed uninfected flies should be included as a control to ensure FAC is not toxic or affecting lifespan independently of infection.

R. In Fig 2e, we included survival of flies fed FAC without infection. None of these flies died. Even after 4 days of continuous feeding on FAC, we did not observe any death, indicating that FAC does not show toxicity within the timeframe that we tested.

Reviewer #2 :

The present study investigates the role of iron sequestration by Transferrin 1 in mediating sex differences to infection. This work follows from a previous study where *Tsf1* was shown to interact with *P. alcalifaciens*. The study shows females succumb to *P. alcalifaciens* to a greater extent, have higher iron loads consistent with *Tsf1* expression levels, and the susceptibility to *P. alcalifaciens* is lost in *Tsf1* mutants, even if sexually dimorphic expression differences are not.

This provides a solid set of initial findings. However, I feel more could be done to provide a conceptual advance, and to shore up the rigour of experiments in this manuscript.

Major comments

1. It's not clear if the loss of sexual dimorphism in *Tsf1* mutants has more to do with a maximum growth rate of the bacteria in a relatively resistance-deficient fly strain. Indeed, if one squints, males still survive slightly longer than females in Fig 3A ($P = 0.11$). The expression of *Tsf1* is sexually dimorphic even in *Rel* and *spz* mutant backgrounds, despite these backgrounds differing in their extent of sexual dimorphism in survival. Overall, this suggests that *Tsf1* mutation causes susceptibility so severe that sexual dimorphism is muted relative to the effect of immune deficiency. But this is not the same as *Tsf1* being causative of the sexual dimorphism. For instance, *Relish* mutants still retain a sexual dimorphism (Fig 4C), but have impaired *Tsf1* induction in both sexes. Fig4B shows *Tsf1* is actually impaired at baseline and induced state in *spz* relative to *Rel*, given the *Tsf1*/RP49 ratio. Showing the sexual dimorphism of Fig3A over a range of doses, or perhaps temperatures to slow the pathogen growth and improve resolution, would confirm if *Tsf1* mutants retain a sexual dimorphism or not.

R. As shown in the graphs below, sexual dimorphism in *Tsf1* mutant was present across different pathogen doses that we tested. Please note that the reduction of pathogen dose did not extend fly survival. It only extended the period till flies started to die. We noticed a similar situation when we tried to reduce the temperature. Hence, typically-used approaches of reducing pathogen dose or temperature do not improve resolution in survival. However, the fact that in another immunocompromised mutant, *Relish*, that is as susceptible to *P. alcalifaciens* as *Tsf1* we still detected the dimorphism, gives us confidence that it is not purely immune deficiency masking the effect of sexual dimorphism in *Tsf* mutant.

2. Following from #1 Attributing the *spz* difference to *Tsf1* is speculative and not demonstrative, particularly given a previous study reported this loss of sexual dimorphism in *spz* mutants was paralleled by *Persephone*, relying on microbial protease sensing and not iron sequestration (Duneau et al., 2017 BMC Biology). Investigations of *Psh* and *Tsf1* alone or combined would strengthen the conclusions here, test the reproducibility of the *Psh* finding (which has not been repeated since 2017, if I'm correct), and provide a more valuable advance for the field.

R. Following the recommendation, we tested the contribution of Psh branch to the sexual dimorphism in survival and *Tsf1* expression (New Figures 4e-g, L285-303). However, we decided to use *hayan-psh* double mutant given their redundant functions in sensing microbial proteases (Dudzic et al., 2019). Consistent with Duneau et al., 2017 who used single *psh* mutant, we found that sex differences in *P. alcalifaciens*-induced Toll activation as measured by *Drs* expression were eliminated (Fig 4e). We also found loss of sex dimorphism in survival in *psh-hayan* mutant (Fig 4f). These results confirm previous findings obtained with *psh* mutant that pathogen sensing by the Psh branch leads to sexually-biased activation of Toll pathway and consequent differences in susceptibility to infection. When we measured *Tsf1* expression, we found that sex differences were still present under both uninfected and infected conditions (Fig 4g). Hence, while *spz* mediates basal differences in *Tsf1* expression between sexes, there is a *hayan*/persephone-independent mechanism of *spz* activation or function that is relevant for basal sex-bias in *Tsf1* regulation.

3. Fig3C: The higher pathogen loads here are taken as an idea that females have higher tolerance to *Providencia* than males (Abstract line 37). But females are larger period, have more hemolymph for bacteria to grow in, and also more iron to promote a faster growth. So can this not simply reflect a higher pathogen load needed to kill females because they're bigger? An argument of tolerance would require showing that the normalized bacterial load per body size (ex: wet mass, dry mass) is somehow different in males vs females. It would be reasonable to say that females increase pathogen virulence by having more iron, but I'm not sure this is really paralleled as "lower tolerance" in the way those terms are used. If nothing more, greater nuance is needed in dissecting the underlying factors behind this result, and I would not simplify it to "lower tolerance," or at least, not without qualifying and justifying what is meant by this. Ideally, some additional body size data could be incorporated into the analysis, which would be relatively quick to do, and improve the ability of the authors to speak to virulence, resistance, and tolerance.

R. The same point was also raised by the other 2 reviewers. We agree that we have insufficient data to fully support the role of *Tsf1* in tolerance. Additionally, inclusion of additional timepoints and reanalysis of log2-transformed data (as suggested by the reviewer) did not reveal statistically significant differences in pathogen load between *Tsf1* male and female flies. Therefore, we removed the claim about tolerance from the manuscript.

4. Line 55, 300: do the authors find females have higher ROS in their settings? In general, an exploration of the melanization reaction and *Tsf1* in these sex differences would improve the study. Presently, the argument is made that *Tsf1* deploys its effect through nutritional immunity, but it could easily be that differences in free iron lead to differences in the potency of the ROS response mediated by melanization, which does not have to do with nutritional immunity. This would provide a more valuable advance for the field.

R. If reviewer's hypothesis "that differences in free iron lead to differences in the potency of the ROS response mediated by melanization" is true, then melanization-deficient mutants should lose dimorphism. Similar to Duneau et al., 2017 who tested *P. rettgeri*, we still detected dimorphism in PPO1,2 mutants (see graph below), suggesting that melanization-derived ROS do not contribute to sex differences in susceptibility to *P. alcalifaciens*. We agree that testing the secondary consequences of sex differences in iron levels is important but would require a separate study.

Minor comments

I appreciate the care to put sample sizes in each figure, which aids the reader to assess the weight of the evidence.

R. Thank you.

Fig1D, Line 200, Fig3C: are the p-values above the boxplots accurate? It's not clear to me how the 6h time point in Fig1D can effectively be $P = 1.00$ with the spread of data points shown. The same comment stands for 6h in Fig3C comparing males, where the cluster of data points hardly overlaps at all, yet somehow the 16h male-female comparisons are significant.

Have the authors analyzed raw values, or log-transformed data? I would recommend analyzing Log2 log-transformed data. i.e. transform the data before running the analysis. Log2 as a transformation because bacteria divide in a 2-fold manner, not a 10-fold manner, so this treatment better reflects a meaningful difference in the log-transformed data in terms of replication cycles achieved within the host. Without this data transformation, differences of thousands of bacteria within a genotype within timepoint (such as the min/max ranges seen in Fig3C), create an artificial skew to the standard deviation and reducing all statistical power. This would explain the binary trend of "significant" or " $P > 0.9999$ " the authors show in boxplots.

R. Following the recommendation, we reanalyzed CFU counts using log2 transformed data and included additional time points as requested by the reviewer 1. Such reanalysis indeed identified significant differences at 6h but not at 16h (updated fig 1d, 3c).

Fig2A and FigS1: it is important the authors demonstrate this *Tsf1* effect does not rely on genetic background. The authors should support the FlyAtlas2 data with survival using Canton S. The authors should also check *Tsf1* expression of OR-R, as they have for the iso line. These quick experiments would strengthen the rigour and conclusions of the manuscript.

R. A similar point was raised by the reviewer 1. We measured *Tsf1* expression and iron content in Oregon R flies and found similar to *w1118* iso phenotypes (Fig S2).

Fig3E, Line 247: It looks like differences in iron load among sexes are present at all time points in Fig3E, even if reduced in magnitude. Strangely, there is not an equal number of data points in each bar, which might be obfuscating some of the differences. For instance, the trend at 15h is quite strong, with no overlap at all, but only marginally significant ($P = .07$) because only 3 data points are

present for Tsf1 in each sex. Moreover, concerns on how the statistics were done for these sorts of plots remain, so I would encourage the authors to 1st) look at the trends and consider the language used in the text, and whether this is justified, independent of what the P-values say. After all one can somewhat modify P-values with justifiable analytical choices (ex: two-way ANOVA vs. glm, or repeated T-tests with FDR correction vs. bonferroni correction, etc...). But the observed consistent trends should not be dismissed simply because within-timepoint comparisons were just off of 'significance'. 2nd) reevaluate the statistical analyses and ensure they are robust. Indeed, the choice of whether to run a whole-data two-way ANOVA with genotype-by-sex, and time point as covariates would likely reveal the interaction term that Tsf1 females have higher iron loads, even if individual time points might fail to reach significance at the current level of sampling within timepoints.

R. Regarding statistics, we think two-way ANOVA is the appropriate choice in this case considering number of variables. We tried running mixed effect model and it returned similar non-significant comparisons for Tsf1 males vs females, hence we leave statistical analysis unchanged.

We agree that Tsf1 females showed the trend toward higher iron level and we acknowledge this in the Results and Discussion sections (Lines 259-260, 357-361).

Line 215: significantly? Please provide the test type, test statistic, and P-value here to confirm what is being discussed. Is this an all-timepoints analysis with timepoint and genotype as covariates? Two-way ANOVA? Glm?

R. These details are provided in the text and figure legends. It is Two-way ANOVA all-timepoints analysis with timepoint and genotype as covariates.

Line 90: clarify grammar

R. Corrected.

Fig3D: are the sample sizes accurate? If so, this experiment must be repeated with 2+ additional replicates as all other experiments have been.

R. We added more repeats.

Line 309: the Toll pathway did not control basal- or infection-based differences in Tsf1 expression, right? Tsf1 remained inducible in spz mutants, and sex differences were consistent, even if lower compared to wild-type.

R. We corrected it. The Toll pathway controls basal but not infection-induced differences as sex differences in Tsf1 expression are gone in *spz* mutant.

Line 330-332: no discussion of Persephone.

R. We added the discussion on psh. L 348-355.

The authors have not made their data available for independent review. The data are available in the manuscript and figures, i.e. not available. Even if the authors just wished to upload Prism files to a FigShare, that would at least allow an independent review to determine the analysis results. For

instance, I could better understand what has gone off in the authors' statistical analyses if the data were available.

R. We provided Excel files in the supplementary material that include data used to make all the graphs.

Reviewer #3 :

In this manuscript, GENETICS-2025-308469, the authors investigated the mechanistic basis of sexual dimorphism in infection outcomes using *Drosophila melanogaster*. The work is focused on nutritional immunity, specifically iron sequestration mediated by Transferrin 1 (Tsf1). The authors showed that males exhibit higher basal and infection-induced expression of Tsf1, resulting in lower iron levels in the hemolymph and improved survival following *Providencia* infection. Importantly, Tsf1 mutants abolish sex differences in survival but retain differences in bacterial load, leading the authors to propose that Tsf1 contributes not only to resistance but also to tolerance. Finally, they demonstrate that sex-biased Tsf1 expression is mediated by the Toll pathway. This is a well-executed study that offers a novel mechanistic insight into sexual dimorphism in immunity via iron sequestration. The work provides an informative contribution to the fields of immunity, host-pathogen interactions, and sexual dimorphism.

R. We thank the reviewer for the favourable evaluation of our work and the constructive suggestions.

The manuscript is clearly written, and the experimental design takes strong advantage of the *Drosophila* system. While some interpretations (particularly regarding sequestering and tolerance) could be better substantiated, the manuscript is of broad interest to geneticists and immunologists. There are the following major and minor points.

Major Points

1. Based on the data presented in Figure 2, the connection between Tsf1 expression (Fig. 2b) and hemolymph iron levels (Fig. 2c) needs clearer interpretation. Males show higher Tsf1 expression both basally and after infection, which should translate into lower hemolymph iron level. This matches the basal data, but after infection iron levels in males and females are comparable. Although, the authors do not seem to compare hemolymph iron levels in male and females at the different time points (0hr, 1.5hr,.....to 16h post-infection), the trend (Fig 2c) shows that females sequester strongly. In other words, the expression and iron measurements do not fully align. This raises the possibility that baseline iron differences, rather than infection-induced sequestration, are the main driver of the survival phenotype. Since, males have high Tsf1 levels and less free iron, so less support for bacteria and more survival. Or there could be additional regulators besides Tsf1 shaping hemolymph iron during infection. The discussion and interpretation should be explicitly revised to reflect this.

R. We substantially revised the Discussion section, which now covers the potential role of additional iron regulators, additional pathways controlling Tsf1, and additional factors underlying sex bias in Toll pathway activity.

2. There is not enough data to support the interpretation of tolerance in Tsf1 mutants. In Tsf1 mutants, sex differences in survival disappear, but females still carry a higher bacterial burden (Fig. 3c). Based on these data, survival and bacterial loads are not aligned; therefore, the idea of tolerance is reasonable but not supported by any data which can include demonstration of reduced

detrimental readouts e.g., ROS; oxidative damage. Please consider either including data or revising to tone down the conclusion.

R. The same point was also raised by the other 2 reviewers. We agree that we have insufficient data to fully support the role of Tsf1 in tolerance. Additionally, inclusion of additional timepoints and reanalysis of log2-transformed data (as suggested by the reviewer 2) did not reveal statistically significant differences in pathogen load between Tsf1 male and female flies. Therefore, we removed the claim about tolerance from the manuscript.

3. The authors showed that overexpressing Tsf1 improves survival in females (potentially by increasing Tsf1 level and subsequently hemolymph iron level) but not in males. The authors explained this as a ceiling effect in males. There is an experimental caveat that the hemolymph iron levels were not shown. Please consider adding data to support the ceiling effect or revise the interpretation and discussion for other explanations, the contribution of other iron regulators such as ferritin or Malvolio.

R. We added to the Discussion the potential role of other iron transporters which could reduce the effect of Tsf1 overexpression in males. L 327-331.

4. The authors showed that Toll pathway activity explains why males express more Tsf1 than females. But the cause of this sex difference in Toll signaling isn't addressed. They briefly mentioned juvenile hormone (JH). Given the major focus of this manuscript on the underlying mechanism of sex dimorphism, it would be important to have a substantial discussion covering other possible mechanisms to fill this gap.

R. We extended the discussion and covered the contribution of additional factors to the sexual dimorphism in the Toll pathway activation, including additional hormones, microbiota differences, and differences in basal level of Toll pathway activity. Lines 370-385

5. The authors demonstrated male-biased Tsf1 effects in two *Providencia* species. That's solid, but it doesn't prove the mechanism is universal. For the broad scope of this manuscript, it would be interesting to know if it is a general principle of nutritional immunity in flies or if it is a pathogen-specific effect that happens to show up in *Providencia*. Please consider adding a discussion segment explicitly addressing whether the same sex-specific effects are expected in other bacterial or microbial infections.

R. A similar recommendation was also made by the reviewer 1. We covered this point in the Discussion, L 399-411

Minor Points

1. In Abstract, please consider introducing Tsf1 as transferrin-1. In the current version, the Tsf1 appears in a manner that assumes prior knowledge and may confuse general readers.

R. Tsf1 is introduced as Transferrin 1 (L32).

2. Please revise for typographical issues. For example, line 152- "the the SYBR Select Master Mix".

R. We carefully proof-read the manuscript.

Please remove duplicate "the". Also in line 157: "on two 10 μ m filter". It should be "filters."

R. Done

3. Please, check Figure legend 3 for italicized gene and genotype names.

R. Corrected.

4. In Figure legends, please define p-value.

R. This was added to the materials and methods.

February 16, 2026

RE: GENETICS-2026-308984

Dear Dr. Iatsenko:

I am pleased to accept your manuscript titled "Sex Bias in Iron Sequestration by Transferrin 1 Modulates Sexually-Dimorphic Infection Outcomes in *Drosophila melanogaster*" for publication in GENETICS, pending minor revisions.

Both reviewers have praised your thorough revisions and do not have any major concerns. As you can see below, one of the reviewers has made a number of suggestions for you to consider, aiming to further improve the manuscript. Please submit your revision along with a brief description of which one of the reviewer's suggestions you were able to address and how. I expect you should be able to submit a revised manuscript within 30 days. A suitably revised manuscript will be officially acceptable for publication.

When revising the ms., please make an effort to shorten it, because that almost always improves a manuscript. We urge authors to heed the advice of Strunk and White: "omit needless words"¹. Follow this link to submit the revised manuscript: Link Not Available

Thank you for submitting this story to Genetics.

Sincerely,

Hamed Jafar-Nejad
Associate Editor
GENETICS

Approved by:
David Greenstein
Senior Editor
GENETICS

Reviewer comments:

Reviewer #2 :

I am satisfied with the authors responses and feel the manuscript is in a good shape. I leave the following comments for consideration before finalization.

R1: This satisfies my concern. I thank the authors for the additional data. It will be good to add this as a supplementary.

R2: I thank the authors for considering this comment, and verifying that there Toll upstream and downstream components contributed independently to the sum phenotype.

Line 294, perhaps the authors should also cite Shen et al. (2023; Sci Adv) who showed in vitro that Hyan does indeed get cleaved by microbial proteases. it is also interesting to learn that deleting Psh-Hyan does not abolish the difference in Tsf1 expression.

Lines 353-355: A minor point, I would be conservative about the language here. The authors did not test Psh alone, but Psh and Hyan. This distinction should be maintained. The authors did not verify the previous Duneau results, but certainly they are consistent (as written in Line 356).

R3: Removing the discussion of tolerance is a fair response, and the present manuscript is clear in its message.

R4: Lines 353-359: regarding the discussion of ROS, it might be worth mentioning that Psh and Hyan are essential for the ROS mediated by the melanization response as an independent mechanism regulated by Toll.

As the authors have PPO1, PPO2 data showing the dimorphism, my comment seems to be incorrect - good to know! If so, it would be good to include this PPO1,2 survival in the supplemental so the field has indications to this point. Without this PPO1,2

data, it is logical to question the disparity of Tsf1 and survival results between spz and Hayan-psh as being reliant on melanization.

Reviewer #3 :

The authors have revised the manuscript entitled "Sex Bias in Iron Sequestration by Transferrin 1 Modulates Sexually-Dimorphic Infection Outcomes in *Drosophila melanogaster*." I have gone through the revised manuscript and the point-by-point response to reviewers.

Overall, the authors have answered constructively and have substantially revised the manuscript. My major comments are addressed either through additional data, clarification, or appropriate revision of interpretations. The revised manuscript satisfactorily addresses my comments, and the conclusions are now adequately supported by the data presented.

Point by point responses to the referees comments on GENETICS-2026-308984 manuscript entitled “Sex Bias in Iron Sequestration by Transferrin 1 Modulates Sexually-Dimorphic Infection Outcomes in *Drosophila melanogaster*”

Reviewer comments:

Reviewer #2 :

I am satisfied with the authors responses and feel the manuscript is in a good shape. I leave the following comments for consideration before finalization.

R. We thank the reviewer for a favourable evaluation of our revised manuscript and valuable suggestions.

We highlighted in yellow text modifications that were made.

R1: This satisfies my concern. I thank the authors for the additional data. It will be good to add this as a supplementary.

R. We have added these data as Supplementary Figure 3.

R2: I thank the authors for considering this comment, and verifying that there Toll upstream and downstream components contributed independently to the sum phenotype.

Line 294, perhaps the authors should also cite Shen et al. (2023; Sci Adv) who showed in vitro that Hyan does indeed get cleaved by microbial proteases. it is also interesting to learn that deleting Psh-Hyan does not abolish the difference in Tsf1 expression.

R. Shan et al. (2023; Sci Adv) was cited.

Lines 353-355: A minor point, I would be conservative about the language here. The authors did not test Psh alone, but Psh and Hyan. This distinction should be maintained. The authors did not verify the previous Duneau results, but certainly they are consistent (as written in Line 356).

R. We changed the phrasing to “consistent with”. L351.

R3: Removing the discussion of tolerance is a fair response, and the present manuscript is clear in its message.

R4: Lines 353-359: regarding the discussion of ROS, it might be worth mentioning that Psh and Hyan are essential for the ROS mediated by the melanization response as an independent mechanism regulated by Toll.

R. We mentioned this in L355-358.

As the authors have PPO1, PPO2 data showing the dimorphism, my comment seems to be incorrect - good to know! If so, it would be good to include this PPO1,2 survival in the supplemental so the field has indications to this point. Without this PPO1,2 data, it is logical to question the disparity of Tsf1 and survival results between spz and Hyan-psh as being reliant on melanization.

R. We have added PPO1, PPO2 data as Supplementary Figure 1f.

Reviewer #3 :

The authors have revised the manuscript entitled "Sex Bias in Iron Sequestration by Transferrin 1 Modulates Sexually-Dimorphic Infection Outcomes in *Drosophila melanogaster*." I have gone through the revised manuscript and the point-by-point response to reviewers.

Overall, the authors have answered constructively and have substantially revised the manuscript. My major comments are addressed either through additional data, clarification, or appropriate revision of interpretations. The revised manuscript satisfactorily addresses my comments, and the conclusions are now adequately supported by the data presented.

R. We thank the reviewer for a favourable evaluation of our revised manuscript.

February 24, 2026

RE: GENETICS-2026-308984R1

Dr. Igor Iatsenko
Max-Planck-Institut für Infektionsbiologie
Genetics of host-microbe interactions
Charitéplatz 1
Berlin, N/A 10117
Germany

Dear Dr. Iatsenko:

Congratulations, your manuscript titled "Sex Bias in Iron Sequestration by Transferrin 1 Modulates Sexually-Dimorphic Infection Outcomes in *Drosophila melanogaster*" is accepted for publication in GENETICS! Many thanks for submitting your research to the journal.

To Proceed to Publication:

1. Format your article according to GENETICS style: <https://academic.oup.com/genetics/pages/author-guidelines>
2. Ensure that you comply with data and community resource citation guidelines:
<https://academic.oup.com/genetics/pages/author-guidelines#section-5-9-2>
3. Upload your final files at <https://genetics.msubmit.net>
4. Add oupsupport@scipris.com and genetics.oup@novatechset.com (or the domains @scipris.com and @novatechset.com) to your email program's "safe senders" list. You will be contacted by both at various points during the production process.

Notes:

- Your currently-accepted manuscript (unedited, as submitted, reviewed, and accepted) will be published at GENETICS and deposited into PubMed as an Advance Access article. Notify sourcefiles@thegsajournals.org before signing your license if you do not wish to publish your article via Advance Access.
- We invite you to submit an original color figure related to your paper for consideration as cover art. Please email your submission to the editorial office or upload it with your final files. You can submit a small-sized image for evaluation, and if selected, the final image must be a TIFF file 2513px wide by 3263px high (8.375 by 10.875 inches; resolution of 600ppi). Please avoid graphs and small type.
- After files are sent to Oxford University Press we use SciPris to manage article licensing and payment. If you do not have a SciPris account, you will receive an email from no-reply@scipris.com to sign up to use Oxford University Press' author portal. After logging in, follow the online instructions to sign your license and arrange any payment due.

If you have any questions or encounter any problems while uploading your accepted manuscript files, please email the editorial office at sourcefiles@thegsajournals.org.

Sincerely,

Hamed Jafar-Nejad
Associate Editor
GENETICS

Approved by:
David Greenstein
Senior Editor
GENETICS

Review comments (if applicable):